# Rapid Protocol Development, Study Startup and Enrolment of a Prospective Study of COVID-19 Vaccination for Patients with Cancer: A Collaborative Approach

**DOI:** 10.3390/vaccines10122003

**Published:** 2022-11-24

**Authors:** Amy Body, Vivienne Milch, Lynda McSorley, Luxi Lal, Elizabeth Ahern, Regina Ryan, Gayle Jones, Dorothy Keefe, Eva Segelov

**Affiliations:** 1Monash Health, Clayton, VIC 3168, Australia; 2Department of Medicine, Nursing and Health Sciences, School of Clinical Sciences at Monash Health, Monash University, Clayton, VIC 3800, Australia; 3Cancer Australia, Surry Hills, NSW 2010, Australia; 4School of Medicine, University of Notre Dame Australia, Sydney, NSW 2007, Australia

**Keywords:** COVID-19, SARS-CoV-2, cancer, study startup, protocol development, vaccination, ethics approval, study design

## Abstract

Background: COVID-19 is an unprecedented global health emergency. It has been highly disruptive for patients with cancer, both due to an increased burden of severe illness and due to pressure on healthcare systems. COVID-19 vaccination has been an important public health measure for this patient group. Aim: The aim of this study was to describe the rapid design and startup of a multicentre study of COVID-19 vaccine response for vulnerable patients with cancer. Study startup: We set up a multicentre prospective observational study of COVID-19 vaccination response for Australian patients with cancer. Due to intensive collaboration between health services, the funding body and laboratories, we were able to develop a protocol and enrol the first patient within 52 days of the initial study proposal. Rapid startup was further enabled by prompt availability of funding and by high-level engagement of institutional review boards, allowing expedited review. Study enrolment: We rapidly enroled more than 500 patients, 80% within 4 months of study opening. Engagement and follow-up were maintained throughout the course of up to five serial vaccination doses. Conclusion: Our study is an example of intensive collaboration inspired by the COVID-19 pandemic and may serve as an example of an agile research response to real-time public health challenges.

## 1. Introduction

In June 2021, an innovative Australian-first prospective clinical study was established to explore the safety and efficacy of SARS-CoV-2 vaccinations in cancer patients. Patients with medical vulnerabilities, including cancer, had largely been excluded from clinical trials measuring the efficacy of SARS-CoV-2 vaccines, with a resultant gap in knowledge about safety and efficacy in this population [1,2,3]. There are several particular concerns about vaccine response in the cancer population, including inherent immune compromise (for example, in haematological malignancies), immune suppression related to treatment (for example, cytotoxic chemotherapy and corticosteroids), and a vulnerable population who frequently have other comorbidities increasing their risk of poor outcome from COVID-19 infection.

At the time of vaccination rollout and study conception in March–April 2021, the Australian population was predominantly naïve to SARS-CoV-2 infection, with a case load of 1181 cases per million population [4]. The population-wide vaccine rollout was later than that in most OECD countries [5]. This created a window of opportunity to study vaccine-induced immunity in a largely unexposed community of patients with all types of cancer, receiving a diverse range of therapies.

The study protocol, named SerOzNET, was developed with consideration of the United States National Cancer Institute (NCI) Serological Sciences Network for COVID-19 (SeroNet) protocol for the surveillance of people with cancer receiving COVID-19 vaccines [6]. SerOzNET was designed to be both “real-world” and flexible. As such, it was able to constantly evolve and add measurements of vaccine efficacy as boosters and new vaccines became available, and importantly as the various SARS-CoV-2 strains including Delta and Omicron became prevalent. The protocol also allowed the addition of new cancer cohorts, including adolescents and children, once vaccination in those age groups received government approval in late 2021 [7].

Australia achieved a high rate of vaccination coverage, with the initial two doses being administered to 95% of the eligible population as of 7 June 2022. However, there has been lower uptake of the third dose (70.1% of eligible population) and subsequent boosters [8]. It is now well known that COVID-19 vaccine-induced immunity wanes relatively quickly over time; therefore, the study of the efficacy of the third, fourth and later doses is critical to inform public health campaigns [9].

The Australian Technical Advisory Group on Immunisation (ATAGI) is responsible for setting vaccine policy with regard to eligibility, including for people with cancer [10]. Recommendations were largely based on expert opinion and frequently updated [11]. Thus, in the setting of the ongoing pandemic and uncertainty regarding the duration of efficacy of COVID-19 vaccines [12], results from our study will inform emerging vaccination recommendations and add robust prospective data to current scientific knowledge. The study aims to provide detailed information regarding serological and cellular vaccine response for immediate and delayed timepoints after serial vaccination (up to five doses as per local recommendations for vulnerable populations).

The aim of this paper is to describe the key enablers and challenges for rapid startup and enrolment in SerOzNET, to contribute to global efforts to improve research efficiency.

## 2. Protocol Development, Study Startup and Enrolment

### 2.1. Protocol Development

In order to capture baseline samples from a large number of participants prior to any vaccination, a rapid startup within this “window of opportunity” was essential. The average study startup time for clinical trials in Australia has been documented to be 159 days, with key delays identified in the ethical review and regulatory processes [13]. A startup time of this duration would have significantly impacted the ability of SerOzNET to enrol participants prior to vaccination.

In February 2021, COVID-19 vaccines became available in Australia and as of 22 March 2021, people with cancer were prioritised to receive vaccination [14]. Cancer Australia is the Australian Government’s national cancer control agency, with a mission including the provision of advice on strategic aspects of cancer care. Recognising the critical role of vaccine protection against COVID-19 for vulnerable Australians with cancer, and the need to build evidence upon which to inform vaccine strategy, Cancer Australia approached cancer centres in Australia in March 2021 requesting expressions of interest to study the short-, medium- and long-term efficacy and side effects of COVID-19 vaccination in cancer patients, based on the NCI SeroNet framework [6].

The Department of Oncology at Monash Health, the largest hospital network in the state of Victoria serving a patient population of over 1 million people, was the successful candidate in the procurement process, and a contract with specific milestones was executed in early June 2021. The detailed study protocol with its adaptive design was then written, including the definition of efficacy and toxicity endpoints [15]. This required a national network of collaborators to be assembled to undertake the specific clinical, qualitative and laboratory analyses in the study. Suppliers for equipment, testing kits and consumables were sourced rapidly, and specific study personnel were employed, including a study manager, lab manager and research assistants.

Flexibility in the date of submission to the Monash Health Human Research Ethics Committee (HREC) was allowed due to the urgent nature of the study, and queries were responded to after the initial meeting without the requirement to return to a second review. Multisite ethics approval for the study was provided on 22 June 2021 (project RES 21-337A). Local governance approval was obtained by individual participating sites in accordance with fully executed collaborative research agreements. The study opened to enrolment at the various sites as listed in the timeline (Figure 1), which shows the startup of the study relative to vaccination policy and rollout.

In total, the time from the initial Cancer Australia procurement approach to the first patient enrolment was only 52 days, reflecting a highly successful and rapid collaborative effort of five institutions and eight different groups (Table 1). Frequent updates to the ATAGI vaccination guidelines required multiple protocol amendments (Table 2). In order to minimise delays, agreement with the HREC and amongst collaborators for rapid amendments within the agile SerOzNET protocol was struck, allowing for adjustments in the number and timing of study material collection related to additional vaccine doses.

A robust governance structure was established, including regular safety monitoring committee meetings and monthly progress reports to Cancer Australia.

### 2.2. Study Database

Use of the Research Electronic Data Capture (REDCap v12.4.22, Vanderbilt University, Nashville, TN, USA) database, a web-based electronic data capture system designed to enable investigators to independently develop and manage electronic databases [23,24], allowed rapid creation and ongoing evolution of a customised SerOzNET database in real time, without potential delays relating to external information technology providers.

### 2.3. Rapid and High-Volume Enrolment

#### 2.3.1. Strategies for Clinician/Referrer Engagement

SerOzNET was championed by a core team of investigators, comprising medical oncologists and haematologists at the lead study site with specific interest in SARS-CoV-2 and associated vaccinations. These champions ensured a high level of engagement from clinicians at the lead site, utilising a dedicated secure encrypted Webex Teams^TM^ channel (Cisco Systems, San Jose, CA, USA) created for internal referrals. This required simple demographic information only and minimal administrative time. A dedicated research assistant and core investigators responded to referrals in real time, communicating with referred patients and scheduling the pre-vaccination visit in parallel with vaccine administration, and transferred patient details to a secure, password-protected database.

The SerOzNET protocol encouraged discussion regarding vaccination with hesitant patients by study investigators. This allowed clinicians with vaccine-hesitant patients to refer their patients for discussion, saving time during routine clinic appointments and facilitating shared decision-making about vaccination.

#### 2.3.2. Patient Engagement—Initial

Patients were referred to the study by their treating physician, and this trust relationship assisted with addressing initial concerns about COVID-19 vaccination. After referral, patients were contacted by the study team within 48 h, maintaining prompt engagement.

SerOzNET commenced during a period of limited vaccine supply in Australia, which resulted in complex statewide booking systems and difficulty accessing early bookings. The study was able to facilitate vaccination at the lead site immediately after sample collection and to book visits on behalf of patients, addressing this barrier to vaccination.

Non-English-speaking patients were contacted via phone interpreters for both the initial approach and all subsequent appointments, allowing the opportunity to discuss any vaccine-related concerns and to ensure full understanding of the study. To date, over one-quarter of adult study participants speak a language other than English at home.

## 3. Ongoing Study Conduct and Patient Engagement

There were a number of enablers which assisted in study conduct and enrolment, including patient engagement strategies and analysis strategies. Conversely, a number of challenges were encountered, such as widespread community transmission at the time of the adolescent vaccination rollout. Table 3 details these enablers and challenges, as well as the responses to challenges.

## 4. Discussion

The international and local collaboration inspired by COVID-19 allowed the SerOzNET study to be rapidly developed. We demonstrated that addressing barriers such as time to ethical review, confirmation of funding and engagement of collaborators allowed the shortening of the startup time from the national average of 159 days (13) to 52 days. Multi-level engagement from a national government agency (Cancer Australia) collaborating with international research institutions (NCI) and subsequently with Australian cancer centres facilitated study conception. Effective engagement between the lead cancer centre and scientific collaborators facilitated rapid protocol development. Timely provision of funding was an important factor allowing expedited startup at the lead site, and conversely, slower contractual approval and subsequent confirmation of funding at other sites was a key reason for the delay in timely secondary site activation.

A streamlined ethics and governance process minimised artificial delays and enabled study approval within a minimum timeframe. Subsequently, the development of an agile protocol in the setting of evolving vaccine recommendations minimised the requirement for repeated revision. Future observational research in the public health setting would benefit from taking this initial approach to protocol development to minimise unnecessary adjustments in response to changing external factors.

Study investigators who were a core part of the target referral group allowed high-level engagement. The ability to offer consultation and discussion to patients regarding vaccination was a bonus of the study for both clinicians and patients as it allowed issues such as vaccine hesitancy to be addressed and provided the opportunity to debunk myths around vaccination. Providing this opportunity for discussion of complex and evolving public health measures increased patient engagement with the study.

Future measures which could improve participant engagement and enrolment include the use of telehealth for initial consultations and consent. The remote nature of telehealth allows clinicians to engage with patients who live at a distance from the clinical site and with those who are hesitant about frequent visits to healthcare facilities during a pandemic. It thus provides an accessible forum for clinician–patient discussion and education regarding the study. Refinement of assays (for example, the use of dried blood spot technology for serological analysis) may improve engagement for groups where blood tests are a barrier to participation (children, difficult venous access). To improve engagement amongst non-English-speaking participants, written translation of patient flyers and information and consent forms could be considered for the most prevalent languages in a catchment area.

This study demonstrates the success of coordinated research efforts and the benefits of early collaboration to deliver a cost-effective and timely service to patients while furthering important public health research. The opportunity to discuss vaccination with patients and provide early information regarding the planned rollout of interventions is a further benefit of this type of investigator-led, adaptive research. The opportunity for early collaboration between the government and key clinical stakeholders allows the preemptive development of protocols to evaluate interventions as they occur and maximise patient enrolment by capturing patient groups from the start of community interventions.

## 5. Conclusions

Very rapid protocol development and study startup are possible with high levels of engagement between national and international funding bodies, clinicians and researchers, and patients and their families and carers. Barriers to study startup were overcome, and our study was commenced more than 100 days faster than the national median startup time. Learnings from this experience can be applied in future clinical research to allow agile research in response to public health challenges. High levels of patient engagement (even requests for additional visits and sample collections) can be maintained by embedding research within an excellent clinical care structure and providing a forum for individual patients to have ongoing discussions of complex and evolving public health recommendations. The unique opportunities and challenges presented by the COVID-19 pandemic and subsequent vaccination rollout will inform clinical research into the future.

## Figures and Tables

**Figure 1 vaccines-10-02003-f001:**
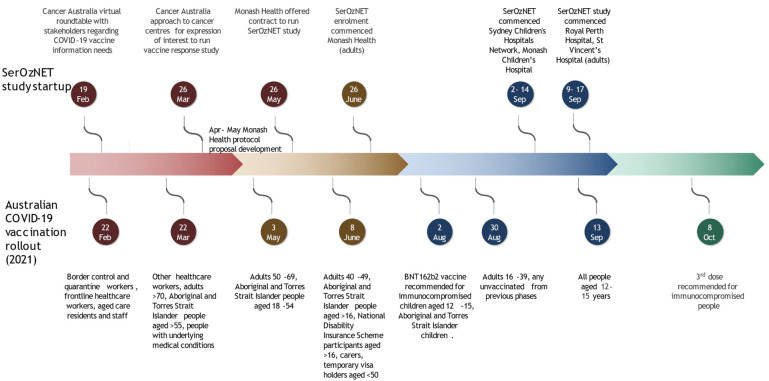
Timeline of Australian COVID-19 vaccine rollout and SerOzNET study startup [16,17,18,19,20,21,22].

**Table 1 vaccines-10-02003-t001:** Collaborating institutions and roles.

Protocol Design Role	Institution
Clinical and statistical protocol development	Monash Health, Victoria
Biostatistics	The Kirby Institute, New South Wales
Neutralising antibody procedures	The Kirby Institute, New South Wales
Peripheral blood mononuclear cell procedures	QIMR Berghofer, Queensland The Kirby Institute, New South Wales
Epigenetic testing	QIMR Berghofer, Queensland
Paediatric clinical design	Sydney Children’s Hospitals Network, New South Wales Monash Health, Victoria

**Table 2 vaccines-10-02003-t002:** Protocol amendments.

Protocol Version	Date	Changes
Version 1	26 May 2021	Original draft
Version 2	8 June 2021	Reduced number of patient visits in response to HREC and consumer representative feedback
Version 3	3 August 2021	Provision for 3rd dose
Version 4	25 August 2021	Provision for inclusion of children aged 12 and over
Version 5	2 September 2021	Minor adjustments during ethical review
Version 6	11 October 2021	Provision for inclusion of children aged 5 and over
Version 7	27 January 2022	Provision for 4th dose, and updated agile framework to incorporate surveillance after potential future additional doses without further protocol amendment

**Table 3 vaccines-10-02003-t003:** Enablers and challenges.

**Enablers**	
*Participant engagement*	
Integration with routine care	Routine blood tests offered at time of appointment Appointments adjusted to coincide with routine care visits or treatments
Provision of emerging vaccine information	Participants were provided phone or in-person updates of relevance to them from ATAGI guidelines, such as eligibility for 3rd, 4th and 5th primary and booster vaccine doses
Provision of information in patient’s language	Routine use of interpreters for non-English-speaking patients
Provision of study updates	Patient newsletters
Avoidance of multiple calls to patients	Use of electronic medical records (e.g., My Health Record [25] to verify healthcare visits during safety monitoring and vaccination doses which may have occurred outside the study site
*Analysis*	
Identification of processing issues	Batched analysis with real-time feedback to participating sites regarding sample quality
**Challenges**	
Slower startup at some sites	Startup was later at some secondary sites due to contractual issues, resulting in delays in enrolment and fewer potentially eligible unvaccinated participants
Community transmission by the time of vaccine approval for adolescents	Rapid rollout of vaccines to young people occurred due to widespread community transmission of the Delta variant at the time of vaccination approval, limiting the number of eligible adolescent participants Addressed prior to enrolment of younger children by protocol update and approval prior to vaccine approval in this age group, to allow immediate enrolment as soon as vaccine was approved
Parental hesitancy for very young children	Blood tests are a concern for parents of very young children, which could potentially be addressed in the future by the development of assays for finger-prick blood testing. Perceived lack of severity of COVID-19 infection for very young children reducing vaccination uptake.

## Data Availability

This project is currently ongoing. After completion, data will be made available on reasonable request.

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
