# Peer review of "Rapid Protocol Development, Study Startup and Enrolment of a Prospective Study of COVID-19 Vaccination for Patients with Cancer: A Collaborative Approach"

_vaccines, 2022, doi:10.3390/vaccines10122003_

Round 1
Reviewer 1 Report
This paper illustrates a multicentre prospective observational study associated with effects of COVID19 vaccination among cancer patients. COVID19 is now a popular topic internationally and in the scientific field, which prompts a lot of scientists to overcome it. They have come up with a fast approach to precipitate the collaboration with others to finish that project. In addition, they also delineate the specific steps of the workflow and indicate the potential issues in each step.
Minor issue:
Figure 1 is not arranged in a proper way and sentences in boxes are not legible due to inappropriate sizes.
More alternative ways should be indicated in terms of how to effectively communicate with non-English-speaking patients.
Author Response
Thank you for your review. I have amended the figure and included further comment in the discussion regarding translation of materials for non-English speaking participants.
Reviewer 2 Report
1) Abstract: L13-23. COVID-19 is an unprecedented global health emergency. It has been highly disruptive for patients with cancer, both due to an increased burden of severe illness, and due to pressure on health care systems. COVID-19 vaccination has been an important public health measure for this patient group. We set up a multicentre prospective observational study of COVID-19 vaccination response for Australian patients with cancer. Due to intensive collaboration between health services, the funding body and laboratories, we were able to develop a protocol and enroll the first patient within 52 days of initial study proposal. Rapid startup was further enabled by prompt availability of funding and by high-level engagement of institutional review boards, allowing expedited review. Subsequently, we rapidly enrolled more than 500 patients and maintained engagement and follow-up throughout up to 5 serial vaccination doses. Our study is an example of intensive collaboration inspired by the COVID-19 pandemic and may serve as an example of an agile research response to real-time public health challenges. Please divide the abstract in different sections (i.e. background, aim, conclusions, …) to clarify it.
2) Introduction. L 59-60. Thus, in the setting of the ongoing pandemic and uncertainty regarding duration of efficacy of COVID-19 vaccines, results from our study will inform emerging vaccination recommendations and add robust prospective data to current scientific knowledge. In order to discuss the previously described points, important references are needed to be added, such as:
a-Interstitial Lung Disease at High Resolution CT after SARS-CoV-2-Related Acute Respiratory Distress Syndrome According to Pulmonary Segmental Anatomy. J Clin Med. 2021;10(17):3985. Published 2021 Sep 2. doi:10.3390/jcm10173985
b- Extinction of the Influenza B Yamagata Line during the COVID Pandemic—Implications for Vaccine Composition. Viruses. 2022; 14(8):1745. https://doi.org/10.3390/v14081745
3) This paper describes the key enablers and challenges to rapid start-up and enrolment in SerOz-NET, and learnings for future research. Please improve the description of study aim.
4) Results. Please underline in the results the data to support the conclusions.
5) 4. Discussion L192-200. International and local collaboration inspired by COVID-19 allowed the SerOzNET study to be rapidly developed. Multi-level engagement from a national government agency (Cancer Australia) collaborating with international research institutions (NCI) and subsequently with Australian cancer centres, facilitated study conception. Effective engagement between the lead cancer centre and scientific collaborators facilitated rapid protocol development. Timely provision of funding was an important factor allowing expedited start-up at the lead site, and conversely, slower contractual approval and subsequent confirmation of funding at other sites was a key reason for the delay in timely secondary site activation. Please summarise here the most important results of the study.
6) 5. Conclusions L230-236. Very rapid protocol development and study start-up is possible with high levels of engagement between national and international funding bodies, clinicians and research-ers and patients and their families and carers. High levels of patient engagement (evenrequesting of additional visits and sample collections) can be maintained by embedding research within an excellent clinical care structure and providing a forum for individual patients to have ongoing discussion of complex and evolving public health recommendations. Please underline the novelty of the study.
Author Response
1) Abstract amended as suggested
2) The suggested references are not relevant to this paper. I have however added an appropriate reference.
3) I have clarified, please see tracked changes.
4) This is a descriptive paper of study startup, not a paper reporting the study results. Therefore, there is not a results section. The write up of results is in process and will be published in due course.
5) As per comment for #4, this paper is a descriptive paper of the startup process. Results will be published separately.
6) Thank you, I have added a clarifying statement in the conclusion.
Reviewer 3 Report
This is a significant study on a prospective study of COVID-19 vaccination.
1. Introduction may be revised to explain the detailed definition of patients with medical vulnerabilities in terms of cancer.
2. Figure 1 may be revised to have the calendar-like timeline format. Otherwise, the dates can be in separate columns or labels.
3. 2. Protocol development, study startup and enrolment may be revised to have Timeline subsection. Each sub-section may be shown with numbers.
4. 3. Ongoing study conduct and patient engagement may be revised to have tables for Enablers and Challenges. Each sub-section may be shown with numbers.
5. Conclusion may be revised to add some description on COVID-19 vaccination.
Author Response
- Done
- Done
- I have instead incorporated aspects of the timeline into Fig 1 for clarity
- I have replaced this section with a table
- I have added a comment
Round 2
Reviewer 1 Report
No big issues are identified.
Author Response
Thank you.
Reviewer 2 Report
The manuscript is interesting but it is quite incomplete.
Rapid protocol development, study startup and enrolment of a prospective study of COVID-19 vaccination for patients with cancer: a collaborative approach is quite interesting but it is incomplete. As reported by authors: This is a descriptive paper of study startup, not a paper reporting the study results. Therefore, there is not a results section. The write up of results is in process and will be published in due course. I think that the paper is incomplete and it is not ready for pubblications.
Author Response
Dear reviewer,
Thanks for your comments.
We have written this paper in response to interest from colleagues at conferences this year regarding the logistics of setting up such a large clinical research project within a matter of short months.
As you can appreciate, this detailed description of study startup is too long to append to an article publishing the study results.
Therefore, we have written a separate paper (this one) which is intended to stand alone.
I understand your concern regarding the lack of results section, however I believe the material in this paper is of use to the clinical research community as a description of our successful startup procedures.
I haven't amended the paper further.
Kind regards,
Amy Body
Reviewer 3 Report
The authors revised the manuscript.
Author Response
Thank you.